# The Mechanism of Facultative Intracellular Parasitism of *Brucella*

**DOI:** 10.3390/ijms22073673

**Published:** 2021-04-01

**Authors:** Hanwei Jiao, Zhixiong Zhou, Bowen Li, Yu Xiao, Mengjuan Li, Hui Zeng, Xiaoyi Guo, Guojing Gu

**Affiliations:** 1Immunology Research Center, Medical Research Institute, Southwest University, Chongqing 402460, China; 2College of Veterinary Medicine, Southwest University, Chongqing 402460, China; zzx449090405@email.swu.edu.cn (Z.Z.); libowenswu@email.swu.edu.cn (B.L.); xiaoyulike@email.swu.edu.cn (Y.X.); lmj123@email.swu.edu.cn (M.L.); zenghui1234@email.swu.edu.cn (H.Z.); greataguo4@email.swu.edu.cn (X.G.); ggj19970819@email.swu.edu.cn (G.G.); 3Veterinary Scientific Engineering Research Center, Chongqing 402460, China

**Keywords:** *Brucella*, facultative intracellular parasitism, immune escape, persistent infection, research progress

## Abstract

Brucellosis is a highly prevalent zoonotic disease characterized by abortion and reproductive dysfunction in pregnant animals. Although the mortality rate of Brucellosis is low, it is harmful to human health, and also seriously affects the development of animal husbandry, tourism and international trade. Brucellosis is caused by *Brucella*, which is a facultative intracellular parasitic bacteria. It mainly forms *Brucella*-containing vacuoles (BCV) in the host cell to avoid the combination with lysosome (Lys), so as to avoid the elimination of it by the host immune system. *Brucella* not only has the ability to resist the phagocytic bactericidal effect, but also can make the host cells form a microenvironment which is conducive to its survival, reproduction and replication, and survive in the host cells for a long time, which eventually leads to the formation of chronic persistent infection. *Brucella* can proliferate and replicate in cells, evade host immune response and induce persistent infection, which are difficult problems in the treatment and prevention of Brucellosis. Therefore, the paper provides a preliminary overview of the facultative intracellular parasitic and immune escape mechanisms of *Brucella*, which provides a theoretical basis for the later study on the pathogenesis of *Brucella*.

## 1. Introduction

Brucellosis is a zoonotic systemic chronic infectious disease caused by *Brucella* [1]. Brucellosis is often called “Mediterranean fever” or “Malta fever”. The main clinical features of the disease are high fever, enlargement of liver and spleen, joint pain, inflammation of reproductive organs and fetal membranes, infertility and localized lesions of various tissues [2,3]. Brucellosis is widely distributed around the world; only a few countries in Northern Europe and Central Europe, as well as Canada, Japan, Australia and New Zealand, have eliminated Brucellosis. The Mediterranean area, Asia and central and South America are the high incidence areas of Brucellosis. Indeed, Brucellosis has been described as being the most common zoonotic disease worldwide with more than 500,000 new human cases annually [1].

*Brucella* is a Gram-negative facultative intracellular parasitic bacteria—it has no spores, flagella and capsules [4]. In 1886, David Bruce first identified and isolated *Brucella* from the spleens of soldiers who died of “Maltese fever” [5]. In 1985, the WHO Brucellosis expert committee divided *Brucella* spp. into six species according to the differences of infected animals and antigenicity. *Brucella melitensis* (*B*. *melitensis*) is the most common and virulent bacteria, followed by *Brucella abortus* (*B*. *abortus*). *Brucella* mainly infects ruminants and causes abortion and infertility in pregnant animals [6]. The *Brucella* cell wall consists of an inner and an outer membrane. The outer membrane of *Brucella* consists of lipopolysaccharide (LPS), outer membrane protein (OMP) and phospholipid layer. Lipid A, core polysaccharide and O antigen constitute the LPS of *Brucella*, which is an important virulence factor of *Brucella* and an important antigen that mediates the immune production of the animal body [7]. *Brucella* mainly invades macrophages and trophoblast cells, parasitizes in host cells through specific molecular mechanism, affects the apoptosis of host cells, thus mediates the autophagy of host cells, and creates favorable conditions for its survival and reproduction in the host cells. After *Brucella* invades the host cells, it mainly causes chronic infection by avoiding the host immune-reaction system [8], but the molecular mechanism of *Brucella* facultative intracellular parasitism has not been clear. Therefore, through an overview of the mechanism of *Brucella* facultative intracellular parasitism, this article provides a theoretical basis for the later excavation of the pathogenic mechanism of *Brucella* and related research.

## 2. Intracellular Life Cycle of *Brucella*

*Brucella* has been initially described as a facultative intracellular parasitic bacteria able to replicate in professional phagocytes such as macrophages, dendritic cells (DC) and granulocytes as well as nonprofessional phagocytes, including epithelial, fibroblastic and trophoblastic cells [9]. *Brucella* interacts with the cell membrane of macrophages through lipid rafts and enters the host cells to form *Brucella*-containing vacuoles (BCV) surrounded by phagocytic vesicles [10]. From 8 to 12 h after *Brucella* invades the cell, BCV obtains some host marker molecules through interaction with lysosome (Lys) and endosomes, matures the endosomes in the membrane bound vacuoles, and forms acidified endosomes. At this time, BCV is called endosomal *Brucella* containing vacuole (eBCV). As BCV develops and matures, the Type IV secretory system (T4SS) mediates the interaction between the effector protein and the endoplasmic reticulum (ER) exit site, and obtains ER and Golgi apparatus-derived membranes. After losing the early host marker molecules, the eBCV obtained Lys marker molecules (such as Rab7, LAMP-1, etc.) [11]. The BCV escaping Lys degradation will reach the ER and fuse with the ER in a Sar1 and Rab2 dependent manner [12]. At this time, the BCV is called repetitive *Brucella* containing vacuole (rBCV). At the later stage of infection, rBCV will be transformed into autophagic *Brucella* containing vacuole (aBCV) (Figure 1). At this time, the aBCV will not continue to mature and kill cells. Thus far, *Brucella* completes the intracellular circulation, and the organism finally releases pathogens through lysis and nonlysis mechanisms [13].

*Brucella* interacts with lipid rafts on the plasma membrane, which promotes *Brucella* to contact with the host cells and mediates its internalization into phagocytes. The lipid rafts contain glycosphingolipids and cholesterol, which can promote the membrane-related biological processes, such as the formation of polybasic membrane complexes, transmembrane signaling and membrane fusion [14]. LPS is a key molecule in the interaction between *Brucella* and host cell lipid rafts, and can prevent complement mediated bacterial lysis and host cell apoptosis [15]. It has been shown that class A scavenger receptor (SR-A) and prion protein (Pr Pc) are involved in the process of *Brucella* invading cells through lipid rafts [16,17]. Prion protein and SR-A, as receptor proteins of heat shock protein 60 (HSP60) and LPS, exist in specific lipid rafts. The destruction of lipid rafts can effectively reduce the early survival of *Brucella* in macrophages, indicating that the introduction of lipid rafts is a necessary condition for the early survival of bacteria [18]. *Brucella* enter the cell to form phagosome and participate in the endocytosis pathway, but *Brucella* can be quickly separated from the phagosome, indicating that the early survival of *Brucella* is related to the lipid raft mediated signaling pathway [19]. In the process of *Brucella*’s intracellular circulation, it needs aBCV to complete the intracellular life cycle and cell–cell diffusion [20]. The host protein (Yip1A) plays an important role in the formation of rBCV and aBCV. In Yip1A knockout cells, *Brucella* could not form rBCV, so it could only remain in the Lys body. The formation of aBCV depends on the small GTP enzyme Rab9 [13]. When the ER Beclin1 and PI3K form a complex, rBCV begins to transform into aBCV, but with the consumption of ATG14L, the formation of aBCV decreases gradually. By interacting with the conserved oligomeric Golgi (COG), the effector protein BspB regulates the COG dependent transport, reorients the golgi body derived vesicles to BCV, promotes the formation of rBCV, and promotes the intracellular proliferation of *Brucella*.

## 3. The Survival and Replication of *Brucella* in Cells

After *Brucella* invades the host cell, it circulates in the cell [21]. The rBCV stage is tightly associated with bacterial proliferation (between 12 and 48 h post infection). At the same time, rBCV obtains a large number of ER molecular markers, such as calmodulin, calreticulin and ER protein Sec61. After *Brucella* infection, unfolded protein response (UPR) is induced by secreting a variety of effector proteins. IRE1 and Yip1A mediate UPR to form a complex at endoplasmic reticulum export sites (ERES) to selfphosphorylate [22]. Activated IRE1 can promote the formation of ER derived vesicles. The ER derived vesicles fuse with the Lys vesicles containing *Brucella* to form rBCV, which promote the proliferation of *Brucella* by continuously fusing with the secretory ER derived vesicles. *Brucella* can survive and propagate in the host cell mainly by the action of LPS, OMP, T4SS, two-component regulatory system and other virulence factors [23,24]. These factors are necessary for *Brucella* to invade host cells and survive and replicate in cells.

As the virulence factor of *Brucella*, the T4SS encoded by VirB can affect the survival and replication of *Brucella* (Figure 2). VirB is a virulence gene of *Brucella*. Its expression in host cells can affect the intracellular survival and replication of *Brucella*. The functions of the 12 genes of VirB operon are different. At present, the effects of the 12 VirB genes on the virulence of *Brucella* are mainly focused on *Brucella. abortus*, *Brucella. melitensis* and *Brucella. ovis* (Table 1).

In the process of *Brucella* cell replication, T4SS plays an important role in a series of processes [30], such as late intracellular and lysozyme markers, the recognition of ER markers and acting on the secretion pathway, the recognition of autophagy markers, resistance to the harsh intracellular environment and the regulation of the activation of the immune pathway [30]. With the help of T4SS, *Brucella* transfers to ER and produces acidic BCV under the induction of expression of VirB operon encoded by T4SS. The acidic environment of BCV is conducive to the survival of *Brucella* [31]. BCV acidification is an important step in the maturation of BCV, and acidic environment contributes to the expression of VirB operon in *Brucella* [32]. *Brucella* induces the expression of VirB operon in acidic environment and controls the expression of genes related to T4SS. *Brucella* uses T4SS to transport effector from the membrane space to the host cell cytoplasm, so as to regulate the signal transduction of the host cell to facilitate its survival in the host cell. In addition to T4SS, the two-component regulatory system (BvrS/BvrR), cyclic β-glucan, Lux R-like transcriptional regulator (VjbR), LPS, flagellumlike structure, and transporter-like protein (Baca) and phosphatidylcholine (PC) are necessary components for *Brucella* to invade cells and live in cells [33]. Cd98hc transmembrane protein also plays an important role in intracellular proliferation and signal pathway regulation [34].

*Brucella* is similar to other intracellular bacteria, it infection of the host cells includes adhesion, invasion, establishment of infection and diffusion. The interaction between host and LPS may play an important role in *Brucella*’s cell viability. The essential gene (manB, wboA) for the O-side chain synthesis of LPS is a necessary factor for *Brucella* to establish the replication region in the cells [35]. Smooth *Brucella* glabrata inhibits host cell apoptosis and promotes its survival and replication in host cells through the interaction of the o-chain and TNF-α [36]. Thus, dead cells do not release specific factors, therefore they do not activate the immune system and *Brucella* are able to avoid host immune surveillance [37]. In this way, *Brucella* enters its replication niche and proliferates in the host cells. In addition, LPS is considered to be the main determinant of virulence, and the survival rate of the LPS-deficient strains in the host cells is reduced. *Brucella* can not only escape from immune surveillance, but also adapt to the internal environment of phagocytes and nonphagocytes and survive in the cell. Based on the inhibition of the phagosomal–lysosomal maturation pathway and the maladjustment of the intracellular transport, *Brucella* enters the cell and then reaches the ER-derived replication niche [38].

The propagation of *Brucella* in cells includes two stages: the stable stage and the exponential stage. The physiological state of the stable stage is favorable for *Brucella* to adapt to the harsh living conditions in the phagocytosis body, while the exponential stage is used to replicate under the appropriate environmental conditions, and this adaptive regulation is completed by molecular mechanism. It has been found that the expression of VirB operon is high in the exponential growth stage, but it is inhibited after entering the stable stage [39]. Like other intracellular bacteria, *Brucella* have adapted to their intracellular lifestyle and no longer need to accumulate energy storage molecules [40]. *Brucella* can survive opsonin mediated phagocytosis and replicate in cells [41]. The presence of cytochromes (such as cytochrome bc1 complex or hydroquinone) with high oxygen affinity play an important role in *Brucella*’s adaptation to intracellular survival. It has been confirmed that *Brucella* uses heme iron polypeptide as an iron source in vitro. When *Brucella* replicates in trophoblast cells, heme iron polypeptide is needed to participate. Macrophages are the preferred host cells of *Brucella*, which are very important for the heme iron polypeptide cycle in mammals [41]. The mutations of iron regulatory genes can weaken the virulence of *Brucella* in mammalian host cells and make it more sensitive to oxidative damage, which indicates that these mutations are involved in the growth and intracellular survival of *Brucella* [42].

## 4. *Brucella* Evades Killing of Host Immune System

Innate immunity and adaptive immunity play an important role in host immunity against *Brucella*. Innate cellular immunity antiinfection effect is not good; the host immune response caused by *Brucella* invasion is mainly adaptive cellular immunity [38].

### 4.1. Brucella Interferes with Innate Immune Recognition and Response of Host

As the first immune defense line, innate immune response plays a very important role in the process of protecting the body from pathogens [43]. *Brucella* has been living in cells for a long time and *Brucella* has evolved many mechanisms that interfere with innate immune recognition and response in its interaction with the body (Figure 3).

The innate immune system recognizes microbes by characteristic molecules like the Gram-negative LPS Lipid A (the LPS bioactive moiety) signals through Toll-like receptors (TLRs) to induce proinflammatory molecules and small GTPases of the p47 family involved in intracellular pathogen control. *Brucella* LPS exhibits a low toxicity and its atypical structure was postulated to delay the host immune response, favouring the establishment of chronic disease. *Brucella* Lipid A is a 2,3-diaminoglucose disaccharide substituted with C16, C18, C28 and other very long acyl chains. This peculiar structure is a poor agonist of TLR4/myeloid differentiation-2 (MD-2) and therefore a paradigm has emerged proposing *Brucella* LPS as a crucial virulence factor that hampers recognition by pattern recognition receptors (PRR) and plays essential roles during infection. Research shows that *Brucella* LPS did not induce inflammatory responses in macrophages and DCs, two of the most important sentinels of the immune system.This was attributed to its poor recognition by TLR4/MD-2, which is widely considered to be the major receptor complex for LPS binding and signalling [44].

Host cells recognize the harmful substances by the interaction of PRR and pathogen associated molecular pattern (PAMPs), so as to stimulate the body to produce related immune response to kill and eliminate pathogens. However, there is increasing evidence that *Brucella* display altered PAMPs in key molecules, suggesting that to escape detection by innate immunity is a survival strategy. One of the best examples of a structure with altered PAMPs is the LPS of *Brucella*. *Brucella* LPS bears a noncanonical lipid A and, although it signals TLR4, it is active only at very high concentrations [45]. The flagellin of *Brucella* lacks the specific domain recognized by TLR5 and plays an important role in immune escape [46]. Moreover, *Brucella* LPS confers a highly resistant phenotype to cationic bactericidal peptides and makes *Brucella* a poor activator of the complement system [47]. The Lipid A of *Brucella* contains a longer fatty acid chain (C28), which greatly reduces its endotoxin properties [23]. There are free hydroxyl residues in the O-chain of bacterial LPS, which is favorable for binding with C3. However, the O-chain of *Brucella* lacks free hydroxyl. Complement C3 can inhibit the production of C3a and C5a by contacting with the specific O-chain of *Brucella* LPS, thus avoiding the capture of the host immune system. At the same time, LPS on the surface of the cell wall has many long side chains, which prevent the membrane attacking complex from contacting the cell membrane (Figure 4).

It was found that the removal of polymorphonuclear before the start of adaptive immunity was helpful to the elimination of bacteria in mice, indicating that neutrophils inhibited the immune response of the body to *Brucella* [48,49]. *Brucella* also has different resistance mechanisms to phospholipase A2, catheteridin, Lys and defensins, so as to ensure its transport in lymphoid tissue. *Brucella* induces antigen-presenting cells to secrete IL-2 and activate NK cells. NK cells secrete IFN-γ, TNF-α, GM-CSF and other cytokines, which play an important role in Th1 and Tc1 reactions [50]. *Brucella* evades the host immune response by affecting macrophage function. *Brucella* can inhibit IFN-γ mediated phagocytosis and TNF-α expression in macrophages [51]. Infected macrophages can produce proinflammatory factors (TNF-α, IL-6, IL-12) and inflammatory chemokines (GRO-α, IL-8, MCP-1). TNF-α can significantly enhance the bactericidal ability of macrophages, IL-12 can induce Th1 immune response and produce IFN-γ. *Brucella* regulates the expression of MHC-I and MHC-II by regulating IFN-γ secretion. IFN-γ mediated Th1 immune response is essential for *Brucella* clearance [52]. *Brucella* can affect the maturation of DC by blocking the TLR2 receptor pathway, and interfere with the establishment of type Th1 immune response by reducing the secretion of IL-12 and preventing the activation of T-lymphocyte by DC [53].

### 4.2. Transmission Mechanism of Brucella Interfering Host Adaptive Immune Response

The adaptive immune response of the organism mainly includes the humoral immune response and the cellular immune response [54]. In the long-term evolution, *Brucella* has produced the transmission mechanism of interfering information from innate immunity to acquired immunity, so as to establish chronic infection [49].

The decrease in macrophage and DCs recruitment after *Brucella* infection leads to the decrease of CD8+ T lymphocyte activation, which leads to the formation of immunosuppression, which is conducive to *Brucella* replication and chronic infection [55]. *Brucella* lumazine synthase (BLS) transmits signals through TLR4 and induces DC maturation and CD8+ T-lymphocytel toxicity, so as to inhibit tumor growth and regulate innate and adaptive immune responses. As one of several virulence factors necessary for *Brucella* to establish chronic infection, prpA can interact with macrophages to promote B-cell proliferation [56]. PrpA can regulate IFN-γ, TNF-α, IL-10 and TGF-β1 in the early stage of infection [57]. The results showed that prpA, Btp1/TcpB and LPS, as immunomodulators, had the ability to inhibit IFN-γ secretion and promote IL-10 secretion, thus affecting Th1 immune response [58].

*Brucella* effectors can control the TLR signaling pathway involved in DC maturation, and have a significant effect on T lymphocyte activation and antigen presentation. The sequence of *Brucella* TIR protein 1 (btp1) is similar to toll/IL-1 domain family, because of the significance of the TIR domain in TLR signal [23,59]. BTP 1 not only inhibited the production of proinflammatory cytokines, but also inhibited the maturation of DC, resulting in the inhibition of TLR2 and TLR4 signals [60].

*Brucella* inhibits immune signal transduction by expressing secreted proteins containing the TIR domain Btp1/TcpB [61]. The detailed mechanism of this protein is still not fully understood, but there is evidence that when it binds to the TIR domain-containing adaptor protein (TIRAP/Mal), it competes with myeloid differentiation response gene 88 (MyD88), which not only promotes TIRAP/Mal ubiquitination degradation but also inhibits TLR4 and TLR2 signal transduction [62]. In this way, *Brucella* inhibits DC maturation and the production of proinflammatory cytokines IL-12 and TNF-α. In addition, this protein also inhibits the killing effect of CD8+ T-lymphocyte on *Brucella* target cells [63].

## 5. Chronic Infection Caused by *Brucella*

After *Brucella* invade the macrophage, with the proliferation of BCV in ER, some components of *Brucella* can interact with host cells, which can inhibit the synthesis of bactericides, affect the activation of signal transduction pathway, induce super allergic reaction, and finally cause systemic persistent infection [64]. When the T4SS encoded by VirB stimulates the body, the immune response produced by the host can inhibit the growth and reproduction of *Brucella*, but it cannot kill *Brucella*, resulting in a confrontation between *Brucella* and the host, causing the body to develop persistent infection [65].

The interaction between *Brucella* and host immune system is very important in causing persistent infection, but immune evasion is not the only mechanism. Studies have shown that the genes needed for *Brucella* persistence are related to changes in bacterial metabolism and the ability of pathogens to utilize specific nutrients [66]. The macrophage is the main target cell of *Brucella* persistence, which provides a place for *Brucella* replication and survival [67]. The key to understanding *Brucella*’s cell viability and chronic infection is to understand the interaction between *Brucella* and different macrophage subsets [68]. Several factors required for the long-term existence of *Brucella* in the host cannot mediate the replication of *Brucella* in in vitro cultured macrophages, indicating that different macrophage populations and their metabolic status may be the determinants of chronic diseases [66,69].

*Brucella* not only has the ability to resist phagocyte sterilization, but also can prevent antigen-specific T-lymphocyte from recognizing themselves, thus forming a microenvironment conducive to their survival and reproduction, leading to chronic persistent infection [28]. The proliferation and replication of *Brucella* weakens the phagocytic function of macrophages, which results in the loss of the cell killing and antigen presenting functions of macrophages, thus avoiding the host’s immune defense mechanism and causing persistent infection [70]. Persistent infection is determined not only by *Brucella*’s ability to evade host immune response, but also by its ability to utilize available nutrients in the chronic phase of infection. Macrophages play an important role in host physiology and metabolism [71], which are the key to the survival of *Brucella*. *Brucella* not only has the ability to resist the phagocytic bactericidal effect, but also makes the host cells form a microenvironment which is conducive to its survival and reproduction. This microenvironment can make *Brucella* live in the host cell for a long time, and eventually lead to chronic persistent infection.

The macrophage is the main target cell of *Brucella*. *Brucella* can inhibit the apoptosis of the macrophage by inhibiting the secretion of TNF-α so that it can survive in the cell and multiply in large numbers, weaken the phagocytic function of macrophage, lose the killing effect and antigenic presenting function of macrophage, and finally escape the immune surveillance of host [72]. On the other hand, macrophages have a weak ability to activate the initial T-lymphocyte and do not express CD1 molecules with antigen presenting function, so they cannot effectively present lipid antigens to NK cells, which may be the reason for the formation of *Brucella* persistent infection. *Brucella* regulates the immune response by inducing regulatory cytokines (such as IL-10), which indicates that the IL-10 pathway plays an important role in the chronic infection caused by *Brucella* [73,74,75].

*Brucella* evades the host’s initial natural immunity through Toll-like receptor (TLR), and causes a slight inflammatory reaction by modifying virulent factors such as LPS and flagellin [51], which causes *Brucella* to develop persistent infection in the body [23]. In addition, *Brucella* can regulate immune response by inducing regulatory cytokines (such as IL-10), indicates that IL-10 pathway can play an important role in bacterial persistent infection [73,74,75]. It is worth noting that *Brucella* IL-10 inhibits the bactericidal ability of macrophages activated by IFN-γ and the production of proinflammatory cytokine in vitro [74,75].

*Brucella* can induce CD4(+) CD25(+) T-lymphocyte to produce anti-inflammatory cytokine IL-10, thus inhibiting the immune activation of macrophages. The early production of IL-10 by CD25 (+) CD4 (+) T-lymphocyte can regulate the function of macrophages and help to promote the initial balance between proinflammatory cytokines and anti-inflammatory cytokines beneficial to pathogens, so as to promote the survival and persistent infection of bacteria [76]. Proline racemase of *Brucella* acts as mitogen of B lymphocytes and induces spleen cells to secrete IL-10, which may be the basis of its persistent infection of mononuclear phagocyte system in mice [77]. In conclusion, these data indicate that IL-10 plays an important role in regulating the initial immune response. *Brucella* infection can increase the survival and persistent infection of pathogens by regulating macrophage function.

Trophoblast cells are also the key target cells for the survival and replication of *Brucella*. The replication ability of *Brucella* in host cells is the core of its pathogenicity. It was found that the accumulation of *Brucella* in trophoblast cells was higher than that in other organs [78]. In the extravillous trophoblast (EVTs), *Brucella abortus* forms an acidified lysosomal vesicle in which it proliferates and replicates. These vesicles play an important role in the chronic infection caused by *Brucella* [79]. It was found that the invasion of *Brucella* to trophoblast cells mainly caused cell necrosis and released a large number of *Brucella*. The necrotic foci of *Brucella* spread continuously through the capillaries and became the main source of chronic infection of the placenta [80].

It has been found that the proliferation of *Brucella* in trophoblast cells is related to the ER network, and *Brucella* establishes a proliferative living environment through the interaction with the endoplasmic network [81]. *Brucella abortus* invades and replicates in the human trophoblastic cell line Swan-71 and that the intracellular survival of the bacterium depends on a functional VirB operon. After *Brucella* infection, the trophoblast cells of ruminants can produce a lot of erythritol, which can promote the growth and reproduction of bacteria. At present, little is known about the changes of the trophoblast environment after *Brucella* infection, which need further research and discovery.

## 6. Conclusions

The relationship between intracellular bacteria and host is very complex, involving a variety of biological factors and signal pathways. As a facultative intracellular bacteria, *Brucella* can colonize, grow and reproduce for a long time after invading the host cell, which is completely dependent on its survival cycle in the host cell and the function of escaping from the host immune system. Although the research on the facultative intracellular parasitism mechanism of *Brucella* is still in progress, there are still many problems to be further explained.

## Figures and Tables

**Figure 1 ijms-22-03673-f001:**
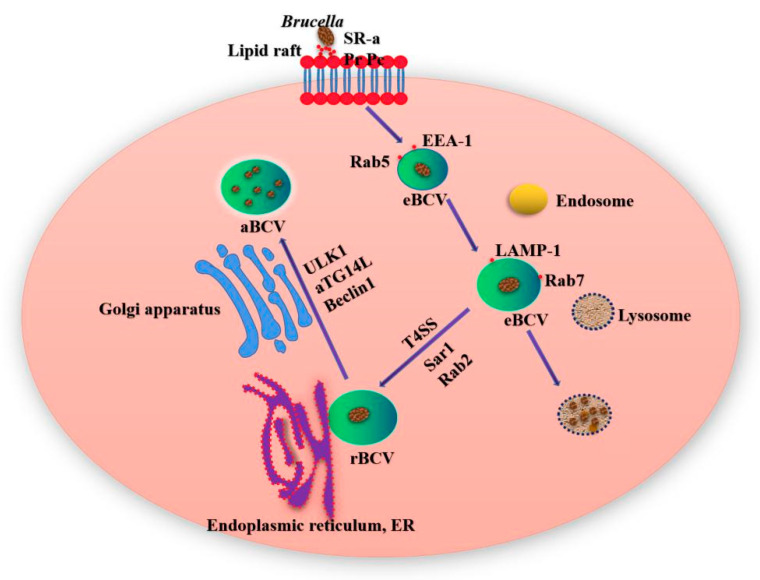
*Brucella* interacts with lipid rafts on the surface of cell membrane to enter macrophages and form *Brucella*-containing vacuoles (BCV). The early BCV in macrophages is called eBCV, it acquires some host marker molecules. With the maturation of eBCV, eBCV loses the marker of early endosome, and obtains the marker molecules of late endosome and lysosome recognition, so as to promote the fusion of eBCV and lysosome. Part of the eBCV escaped lysosome degradation and reached the ER, and then fused with the ER by Sar1 and Rab2 to form rBCV. *Brucella* proliferated in rBCV. At the late stage of infection, the rBCV containing a large number of *Brucella* transformed into aBCV. The aBCV released pathogens through cleavage and noncleavage mechanisms, and the intracellular circulation of *Brucella* ended.

**Figure 2 ijms-22-03673-f002:**
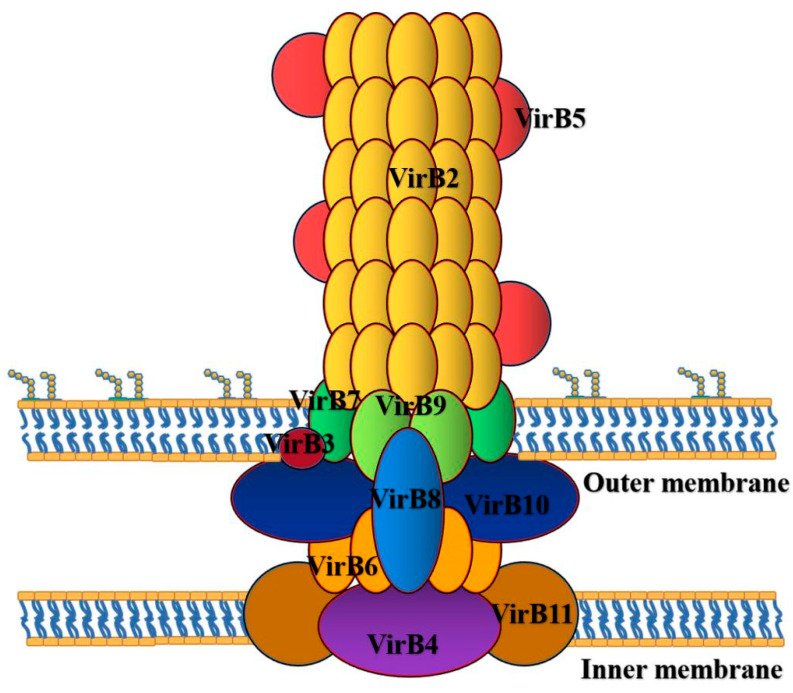
T4SS of *Brucella* is a multiprotein complex encoded by VirB operon, which participates in the intracellular activities of *Brucella*. The T4SS is mainly divided into the following five parts: the elongation region is composed of VirB2; the central and outer membrane regions were composed of VirB7, VirB9 and VirB10; the junction region is composed of VirB5 and VirB10; the intimal region was composed of VirB3, VirB4, VirB6, VirB8 and VirB10; ATP energy region composed of VirB4 and VirB11.

**Figure 3 ijms-22-03673-f003:**
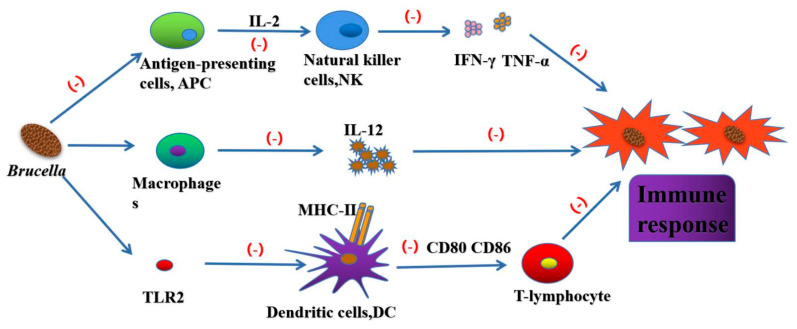
*Brucella* can inhibit the secretion of IL-2 by antigen presenting cells, and then prevent natural killer cells (NK) from secreting inflammatory factors such as IFN-γ and TNF-α. *Brucella* can also inhibit IFN-γ mediated phagocytosis to escape the killing of immune system. *Brucella* affects the maturation of DC by blocking TLR2 receptor pathway, and interferes with the establishment of Th1 immune response by inhibiting macrophages to reduce IL-12 secretion and preventing DC from activating T-lymphocytes.

**Figure 4 ijms-22-03673-f004:**
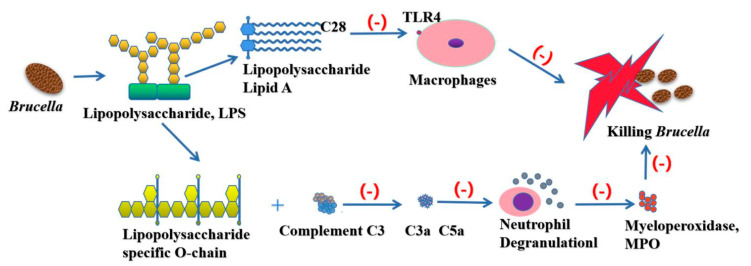
As an important virulence factor of *Brucella*, LPS plays an important role in escaping from the host immune system. The acetyl side chain (C28) of *Brucella* LPS reduces the nature of endotoxin to avoid the recognition of Toll-like receptor 4 (TLR4), thus avoiding the monitoring of host immune system. The specific O-chain contained in *Brucella* LPS can inhibit the production of C3a and C5a by complement C3, and then inhibit the degranulation of neutrophils, so as to prevent the release of myeloperoxidase (MPO) and other lysosomal substances, and prevent them from being captured by the host immune system.

**Table 1 ijms-22-03673-t001:** The 12 genes of VirB operon and their respective function *Brucella. abortus*, *Brucella*. *melitensis* and *Brucella. Ovis* [25,26,27,28,29].

Operon	Whether or Not the Virulence Decreases after Deletion			Function
	*Brucella. abortus*	*Brucella. melitensis*	*Brucella. ovis*	
VirB1	decrease	unknown	unknown	Dissolves glycosyltransferase, which makes T4SS easier to assemble and assemble.
VirB2	decrease	decrease	decrease	Participate in the immune protection of the body and affect the production of immune protection.
VirB3, VirB6, VirB7, VirB10	decrease	unknown	unknown	Signal transmission of bacterial transmembrane proteins.
VirB4	decrease	decrease	unknown	Transport substances and effectively prevent BCV and dissolution Enzyme fusion.
VirB5, VirB8	decrease	unknown	unknown	Regulation of intracellular transport in *Brucella*.
VirB9	decrease	decrease	unknown	As a part of the outer membrane of Brucella type IV secretion system, it can stimulate the body to produce immune responsean.
VirB11	decrease	unknown	unknown	It provides energy for the secretion process of *Brucella*.
VirB12	unchanged	unchanged	unchanged	Immune antigen.

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
