# Peer review of "The Mechanism of Facultative Intracellular Parasitism of Brucella"

_ijms, 2021, doi:10.3390/ijms22073673_

Round 1

Reviewer 1 Report

The Authors describe "The mechanism of intracellular parasitism of Brucella”.

The topic is very interesting and the manuscript is well written, but in my opinion, from a scientific point of view, the text doesn’t offer anything new. Furthermore in the introduction section there are some inaccuracies about the main features of Brucella.

In fact:

  • At the page 2, line 3, in the keywords and in the title also, the Authors said that Brucella is an “intracellular parasitic bacteria”;
  • At the page 2, line 4, the Authors said also that Brucella “has capsules”.

Not only isn’t Brucella an intracellular (obliged) parasitic bacteria, but also it doesn’t have a capsule.

All members of Brucella genus are “FACULTATIVE” intracellular bacteria, without a capsule; therefore, if I were the Author, I’d change the title, adding FACULTATIVE intracellular parasitism, in order to avoid confusion to readers.

  • At the page 2, line 9 the Authors said ……” Brucella bovis”……..

What is that? Brucella bovis hasn’t been discovered yet!!

Did the Authors mean “Brucella abortus”?

However, as it is a review, looking at it from a tutorial perspective, as far as I’m concerned, the paper could be published, but the Authors should correct, at least, the inaccuracies highlighted in the introduction section.

Author Response

Point 1: At the page 2, line 3, in the keywords and in the title also, the Authors said that Brucella is an “intracellular parasitic bacteria”;

At the page 2, line 4, the Authors said also that Brucella “has capsules”.

Not only isn’t Brucella an intracellular (obliged) parasitic bacteria, but also it doesn’t have a capsule.

Response 1: As you suggested, we have made  changes in line 38 and line 39 in the revised manuscript. We thank you very much for your advice!

Point 2: All members of Brucella genus are “FACULTATIVE” intracellular bacteria, without a capsule; therefore, if I were the Author, I’d change the title, adding FACULTATIVE intracellular parasitism, in order to avoid confusion to readers.

Response 2: As you suggested, we have made  changes in line 2, line 24, line 38, line 50, line 56, line 60, line 377 and line 380 in the revised manuscript.

Point 3: At the page 2, line 9 the Authors said ……” Brucella bovis”……..

What is that? Brucella bovis hasn’t been discovered yet!!

Did the Authors mean “Brucella abortus”?

Response 3: As you suggested, we have changed "Brucella bovis" to "Brucella abortus" at line 43 of the revised manuscript.

Reviewer 2 Report

This is an interesting topic and merits attention. I have proposed some areas of revision in the attached PDF. Specifically, more clarification of certain mechanisms described throughout the manuscript and their significance. On the other hand, I felt that there were areas of repetition that could be omitted and other areas that should be re-arranged. 

I attach comments on the PFD file. 

Author Response

We are very grateful for your constructive comments. We have revised your comments very carefully. Please refer to the red part of the revised manuscript for details.